# Revealing the Heterogeneity of the Tumor Ecosystem of Cholangiocarcinoma through Single-Cell Transcriptomics

**DOI:** 10.3390/cells12060862

**Published:** 2023-03-10

**Authors:** Jihye L. Golino, Xin Wang, Hoyoung M. Maeng, Changqing Xie

**Affiliations:** 1Thoracic and GI Malignancies Branch, Center for Cancer Research, National Cancer Institute, National Institutes of Health, Bethesda, MD 20814, USA; leej64@mail.nih.gov (J.L.G.); wangx49@mail.nih.gov (X.W.); 2Vaccine Branch, Center for Cancer Research, National Cancer Institute, National Institutes of Health, Bethesda, MD 20814, USA; hoyoung.maeng@nih.gov

**Keywords:** cholangiocarcinoma, single-cell RNA sequencing, heterogeneity

## Abstract

The prognosis of cholangiocarcinoma remains poor. The heterogeneity of the tumor ecosystem of cholangiocarcinoma plays a critical role in tumorigenesis and therapeutic resistance, thereby affecting the clinical outcome of patients with cholangiocarcinoma. Recent progress in single-cell RNA sequencing (scRNA-seq) has enabled detailed characterization of intratumoral stromal and malignant cells, which has vastly improved our understanding of the heterogeneity of various cell components in the tumor ecosystem of cholangiocarcinoma. It also provides an unprecedented view of the phenotypical and functional diversity in tumor and stromal cells including infiltrating immune cells. This review focuses on examining tumor heterogeneity and the interaction between various cellular components in the tumor ecosystem of cholangiocarcinoma derived from an scRNA-seq dataset, discussing limitations in current studies, and proposing future directions along with potential clinical applications.

## 1. Introduction

Cholangiocarcinoma (CCA) is a highly lethal malignancy originating from the bile ducts, and the global incidence of CCA has been rising over the past few decades [1,2]. This type of cancer can be classified into intrahepatic (iCCA) and extrahepatic CCA (eCCA) depending on the anatomical location. It has been found that the anatomical subtypes of CCA display unique molecular aberrations [3] that suggest complexities in the pathogenesis of CCA and contribute to the heterogeneity of the tumor ecosystem (TE). Curative-intent surgery with and without adjuvant capecitabine is standard-of-care treatment for early stage CCA, and has a median overall survival (mOS) of 49.6 months and 36.1 months, respectively [4]. Recent progression with liver transplantation in patients with eCCA has brought hope to this malignant disease [5,6]. Moreover, a prospective study reported a 5-year survival of 83% for selected patients with local advanced iCCA who underwent liver transplantation followed by neoadjuvant chemotherapy [7]. These studies have increased interest in liver transplants as a form of treatment for patients with CCA [8,9,10].

However, most patients are diagnosed with advanced disease and have limited available therapeutic options [11]. In the last decade, gemcitabine plus platinum-based chemotherapy has been the only first line therapy for patients with unresectable CCA. Numerous clinical trials have failed to improve the OS in patients with unresectable CCA using gemcitabine/cisplatin combination as the main treatment [12], including the recently reported SWOG 1815 trial [13]. In 2022, anti-programmed cell death ligand 1 (PD-L1) durvalumab in combination with chemotherapy received FDA approval as a viable first line therapy based on the TOPAZ-1 trial, which showed an objective response rate of 26.7% and mOS of 12.8 months [14]. In the second line setting, other than commonly used chemotherapeutic drugs (e.g., FOLFOX regimen [15] and liposomal irinotecan/5-FU [16]), there are several targeted therapeutic agents that have been approved for a select few cases bearing specific molecular aberrations. These targeted agents include pemigatinib [17], infigratinib [18], and futibatinib [19] for FGFR2 fusion/rearrangement and ivosidenib [20] for IDH1 mutation CCA patients. Although there has been rapid expansion of available treatment options for this aggressive malignant disease, the efficacy and response rate from such therapeutic combinations have been suboptimal. Therefore, novel treatment strategies for this lethal tumor are urgently needed, and improvement in our understanding of the pathogenesis of CCA to inform the basis of therapeutic strategies could facilitate the exploration of new therapies [21].

The tumor ecosystem (TE) is an environment consisting of cancer stem cells (CSCs), tumor cells, stromal cells, extracellular matrix, and a broad spectrum of signaling molecules present in the TE [22]. It is a micro-battlefield where different groups of players exchange signals, orchestrate tumorigenesis, and respond to immune signals and therapy. The availability of global transcriptomics derived from bulk CCA tumor samples has opened up a new path to decipher the TE of CCA, which leads to the categorization of CCA into different molecular subclasses [23]. However, even with the development of algorithms to interpret immune cell composition and characteristics, the analysis of bulk samples provides information with limited resolution in terms of the composition of different cell populations and their interaction with the different cell types within the tumor [24,25,26].

The introduction of single-cell RNA sequencing (scRNA-seq) provides extraordinary opportunities to investigate transcriptomics at the single-cell level. This technology includes several steps, such as tissue/organ dissociation, single-cell capture, cell lysis, RNA extraction, cDNA synthesis, library construction, single-cell sequencing, and data analysis. This platform has enabled large-scale and high-resolution profiling of transcriptomic states/stochasticity for a deeper understanding of the diversity of cell states and the heterogeneity of cell populations that reside within the tumor. It also allows researchers to identify different cell types, including rare cell populations, discover novel pathological processes, and facilitate the exploration of new effective therapeutic regimens. Here, we review findings about the TE of CCA from available research with single-cell transcriptomics (Table 1) based on 10× genomics sequencing platforms and how this information can be applied to improve future therapeutic approaches.

## 2. Heterogeneity and Plasticity of CCA Tumor Cells

Tumor cells are usually made up of functional heterogeneous populations that are in different cellular states and may change over time. There is growing evidence indicating that intra-tumoral heterogeneity in primary tumors is not only determined by the genetic and epigenetic features of cancer cells but can also be influenced by various cues present in the TE.

Previous studies that have carried out scRNA-seq analysis on iCCA tumor cells have found high degrees of intertumoral heterogeneity [27,28]. The tumor cells of iCCA shared common activated signaling pathways, including IL-6/STAT3, Wnt, transforming growth factor (TGF), and tumor necrosis factor (TNF) [28]. However, four different subsets of tumor cells with distinct transcriptomic patterns based on different gene signatures including epithelial–mesenchymal transition (EMT), cell-cycle and hypoxia, interferon response, and high levels of serine peptidase inhibitor Kazal type 1 (SPINK1) were identified. The different signatures were controlled by distinct underlying transcription factors. In the study, SPINK1, a kinase inhibitor of premature trypsin activation, was found to be associated with cancer stemness and poor prognosis [28]. These results indicate the necessity of developing multi-target strategies in order to eradicate the heterogenous population of CCA tumor cells. When taking histological classification of iCCA tumors into consideration, scRNA-seq data of iCCA samples can be incorporated to further understand tumor diversity. Based on 14 pairs of iCCA tumors and non-tumor liver tissue samples, these iCCA tumors can be divided into S100P+SPP1− and S100P−SPP1+ groups, which not only represent two different origins of iCCA, one from the peripheral large and another from the small duct, but also different biological functionals. The S100P+SPP1− iCCA is significantly associated with worse prognosis in comparison to S100P−SPP1+ iCCA [30]. Even among S100P−SPP1+ iCCA cells, there were two distinct groups that exclusively expressed ALB or ID3, which represent the differences in differentiation status and stemness of iCCA cells, respectively. ID3 expression was correlated with cancer associated fibroblasts (CAFs) and poor prognosis [30], which is consistent with the theory that cancer stemness is associated with poor disease outcome. Additionally, with different murine models, two different transcriptomic subtypes of iCCA malignant cells were identified. One cell type is AP-1 positive with a high expression of growth receptor genes, Fgfr2 and Igfr1, as well as AP-1 target genes, Jun and Fos. This cell subgroup was annotated as a “stress-responding” subgroup. The other cell type expressed markers for cell proliferation, Mki67 and Cdk1, and was referred to as a “proliferating” subgroup. The findings were confirmed with a human scRNAseq dataset [32]. The results indicate the necessity of combined therapy for improved patient outcomes.

The heterogeneity of tumor cells can be quantitatively measured with a transcriptomic diversity score based on scRNA-seq data [27]. Among different types of liver cancer, including hepatocellular carcinoma (HCC) and CCA, high levels of transcriptomic diversity tend to be associated with more aggressive tumors and worse overall survival. Tumor diversity seems to be triggered by hypoxia, evidenced by a high positive correlation between tumor diversity and hypoxia-induced genes, including hypoxia-inducible factor 1α (HIF1a) and the direct downstream target, vascular endothelial growth factor A (VEGFA). The transcriptomic diversity within a tumor affects T cell activity, where T cells from higher heterogeneous tumors showed lower cytolytic activity, which may be used to predict treatment response to immunotherapy [27]. These findings suggest the potential development of therapeutic strategies to combine anti-VEGF agents with other modalities in advanced CCA. The heterogeneity of CCA tumors can also be evaluated by tumor cell transcriptome-based functional clonality [33]. The increased functional clonality was accompanied by an increase in proliferative pre-exhausted T cells, where osteopontin may play a key role in tumor cell evolution and subsequent reprograming of the TE, resulting in overall worse prognosis [29].

Cancer stem cells (CSCs) have been recognized in various tumor types to be responsible for long-term maintenance of tumors and thought to play a role in tumor initiation, tumor recurrence, metastasis, and treatment resistance. The interaction between cancer stemness and immunogenicity of CSCs in the iCCA is largely unknown. Our group used publicly available scRNA-seq data [27,28] to study the stemness of malignant cells in human iCCA [28]. Using an established computerized method, CytoTRACE, we found significant heterogeneity in stemness/differentiation states among iCCA malignant cells [34]. We demonstrated that malignant cells with high stemness expressed much lower levels of major histocompatibility complex II molecules when compared to low stemness malignant cells, suggesting that high stemness malignant cells have an intrinsic mechanism for immune evasion. In addition, high stemness malignant iCCA cells exhibited significant expression of certain cytokines, including CCL2, CCL20, CXCL1, CXCL2, CXCL6, CXCL8, TNFRSF12A, and IL6ST, indicating proactive communication with surrounding immune cells. These results suggest that tumor cell plasticity helps high stemness malignant cells to retain their intrinsic immunological features and also facilitates the escape of immune surveillance [34].

## 3. Heterogeneity of CCA Cancer-Associated Fibroblasts (CAFs)

CAFs are one of the major stromal cell populations within the TE of various solid tumors and play a fundamental role in tumorigenesis, therapy resistance, and survival outcome. One of the histological hallmarks of CCA is the presence of a profuse stroma, which contains diverse CAF populations and an abundant matrix generated by CAFs. This has resulted in great interest in dissecting these cells in CCA. Emerging evidence from genetic profiling [35,36] to protein [35,36,37] data analysis has indicated distinguished roles for CAFs in CCA. These cells have been found to facilitate intense desmoplastic organization and rearrangement of the extracellular matrix (ECM) in the TE of CCA. Single cell transcriptomics from CCA provide further elucidation on the heterogenous CAF population in the TE.

With *Lrat*-Cre-driven lox-stop-lox-*TdTomato* to label hepatic stellate cells (HSCs), as well as the depletion of HSCs, it was found that the majority of CAFs in four mouse iCCA models were derived from HSCs. This observation was further confirmed with scRNA-seq, where the majority of CAFs shared the same expression of both a CAF and HSC signature. Moreover, cell–cell interaction analysis based on scRNA-seq from human and mouse iCCA samples indicated that HSC-derived CAFs dominated the interaction with tumor cells. This interaction promoted tumor growth [31]. Notably, the CAF population showed transcriptomic heterogeneity and could be separated into five subsets, which include inflammatory and growth factor enriched CAFs (iCAFs), myofibroblastic CAFs (myCAFs), mesothelial CAFs (mCAFs), multiple-category CAFs (multi-CAFs), and other CAFs. Among them, MyCAFs (Col1al^+^ and SERPINF1^+^) had upregulated expression of extracellular matrix pathways and were associated with tumor cell proliferation, intraneural invasion, decreased survival, and higher tumor recurrence. MyCAFs were found to promote CCA tumor growth through hyaluronan synthase 2/hyaluronan (rather than type I collagen) mediated by the interactions with non-tumor cells and tumor cells. The iCAFs (Rgs5^+^) subset expressed high levels of quiescence markers and was enriched for inflammation, growth factors, and antigen-presentation genes, as well as various cytokine activity pathways. It was found that iCAF-derived hepatocyte growth factors promote CCA tumor growth through interaction with tumor cells, inducing expression of MET [31].

Through a negative selection strategy (EpCAM^−^CD45^−^CD31^−^ cells), over 2000 CAFs from two human iCCA samples were analyzed with scRNA-seq and shown to have transcriptomic heterogeneity. These subsets were annotated based on their transcriptomic features: vascular CAFs (vCAFs), matrix CAFs (mCAFs), inflammatory CAFs (iCAFs), antigen-presenting CAFs (apCAFs), and epithelial–mesenchymal transition-like CAFs (eCAFs). Among these subsets, vCAFs was the most abundant type (57.6%) and was characterized by the presence of microvasculature, proliferation signature genes, and highly expressed inflammatory chemokines, including IL6 and CCL8. These CD146+ vCAFs are mainly located in the tumor core and microvascular region. Therefore, vCAFs are considered to be connected to the tumor lymphatic vasculature given their gene expression pattern and location in the tumor [38]. Nevertheless, vCAFs were found to actively interact with tumor cells via the pro-invasive IL6/IL6R axis. This axis functions to enhance epigenetic modification of tumor cells, which subsequently promotes cancer stemness and contributes to iCCA progression. On the other hand, CCA tumor cells augment IL6 production in vCAFs through the secretion of exosomes containing miR-9-5p to form a regulatory loop between tumor cells and vCAFs. In this study, mCAFs (POSTN+) were found to be present in the invasive front of the tumor, primarily located within collagen-rich stromal streaks, indicating their association with tumor invasion [28].

In addition, the CAF population of human iCCA in another study was characterized into four different subsets including HSPA1A+, ID4+, THBS2+, and THY1+ CAFs, further confirming the heterogeneity of CAFs in iCCA. The frequency of these four subsets of CAFs was different depending on tumor clonality. For example, HSPA1A+ and ID4+ cells are enriched in liver tumors with less tumor clonality. Interestingly, none of these studies provided a detailed list of differentially expressed genes used to annotate different CAF subsets. It would be worth revisiting all the available data and creating a consensus among the different annotations.

## 4. Heterogeneity of CCA Immune Cells

### 4.1. Lymphoid Compartment

Tumor infiltrating immune cells are associated with CCA prognosis [39,40]. Because of their ability to recognize mutations in tumor cells, directly mediate cancer cell death, as well as being one of main tumor infiltrating lymphocytes, T cells are at the center of cancer immunology and have been considered as the main target in cancer immunotherapy. The scRNA-seq study of CCA has shown that infiltrating T cells were enriched with a plethora of signaling pathways, including increased hypoxia, apoptosis, and IFN response, alongside decreased oxidative phosphorylation [28]. It was noted that intra-tumoral heterogeneity, which is measured by a tumor diversity score, seems to affect the overall phenotype of T cell populations in CCA [27]. T cells derived from highly diverse tumors were mainly enriched in the epithelial–mesenchymal transition and myogenesis pathways, whereas T cells derived from tumors with low diversity were enriched in allograft rejection, oxidative phosphorylation, fatty acid oxidation, interferon (IFN)-a/IFN-g response, MYC activity, and proliferation pathways. These findings indicate that these CCA T cells may have different cytolytic anti-tumor activities and distinct metabolic features which may be decided by tumor diversity. The presence of these pathways may serve as an indicator for tumor immune surveillance status but also as biomarkers for predicting responses to immunotherapy [27].

Diverse functional/phenotypic T cells have been further elaborated in CCA at the single-cell level [27,28,33]. Among them, CD8 T cell subpopulations expressed different levels of cytotoxic markers such as granzyme A (*GZMA*), *GZMB*, *GZMK*, perforin (*PRF1*), and *IFNγ*, indicating various level of cytotoxic activity within this cell population [27,28,33]. Cytotoxicity-related genes in CD8 T cells (e.g., GZM family and *PRF1*) were upregulated in diversity-low tumors compared to those in diversity-high tumors [27], implying that diversity-low tumors may be more responsive to immune checkpoint inhibition therapies. Interestingly, the proliferating CD8 T cells were found to express certain exhaustion markers, such as lymphocyte-activation gene 3 protein (*LAG3*), T cell immunoreceptor with Ig and ITIM domains (*TIGIT*), and T cell immunoglobulin mucin receptor 3 (*TIM3*, or *HAVCR2*), suggesting that these cells were exhausted despite their status as conventionally growing cells. However, immune checkpoint molecules were not only expressed in exhausted CD8 T cells (CD8+PDCD1+) but also in pre-exhausted CD8 T cells (CD8+CXCL13+), indicating another potential target for immunotherapy. It was found that tumor cell clonality is associated with polarization of the T cell landscape [33]. Large proportions of memory CD8 T cells (CD8+GZMK+, and CD8+IL7R+) as well as a group of cytotoxic CD8 T cells (CD8+GNLY+) were enriched in liver cancers with low clonality. In contrast, proliferative pre-exhausted CD8+MKI67+CXCL13+) and regular pre-exhausted T cells (CD8+CXCL13+) were enriched in liver cancers with high clonality. Furthermore, levels of cytokines and chemokines were much higher in CD8+ T cells in liver cancers with low clonality in comparison to tumors with high clonality. Cytotoxic CD8 T cells (CD8+GNLY+) were found to be a major source of cytokines and chemokines secretion in liver cancers with low clonality, while proliferative pre-exhaustion T cells (CD8+MKI67+CXCL13+) were the main source in liver cancers with high clonality. In addition, CD8 T cells in the liver tumor with low clonality were enriched in immune response-related pathways, which were not found in CD8 T cells from liver cancer with high clonality. The analysis of ligand–receptor interactions showed much stronger interactions between malignant cells and CD8 T cells in liver cancer with high clonality than low clonality, especially the ligand–receptor pair SPP1–CD44. These results support the key role of SPP1 in the TE, and suggest that blocking the SPP1–CD44 axis may affect the interaction between CD8 T cells and malignant cells and serve as a potential therapeutic strategy for CCA.

There is a similar effect of tumor cell clonality on CD4 T cell polarization [33]. It was noted that memory CD4 T cells (CD4+CD69+) were enriched in liver cancers with low clonality, while proliferative pre-exhausted CD4 T cells (CD4+MKI67+CXCL13+) were enriched liver cancers with high clonality and a major source of cytokines and chemokines. In addition, CD4 T cells in the liver tumor with low clonality were enriched in immune response-related pathways, whereas high clonality malignant cells had weaker ligand–receptor interactions with immune cells [33]. Several heterogenous non-Treg CD4 T cell populations were also explored, including CD4+IL7R+, CD4+CD27+, CD4+GZMA+, CD4+CTLA4+, CD4+ANXA1+, and CD4+CD69 cells [27]. Among them, CD4+CD69+, CD4+ANXA1+, CD4+CTLA4+, and CD4+GZMA+ were associated with diversity-high tumors, suggesting a correlation between CD4 T cell polarization and tumor diversity.

Tregs are part of the immune population with highly immunosuppressive characteristics and mostly localized to the peritumoral region. They execute immunosuppressive functions through secreting a plethora of inflammatory cytokines, including IL-10 [41,42] and TGF-β1 [43], as well as metabolizing extracellular ATP to adenosine [44,45], which subsequently diminishes the antitumor activity of NK and cytotoxic CD8+ T cells. Moreover, overexpression of the transcription factor Foxp3, a distinct marker of their immunophenotype, upregulates CTLA-4, which inhibits CD8+ T cell activation by binding to CD80 expressed by antigen-presenting cells. It was found that Tregs in CCA express genes of CTLA-4, TIGIT, and TNFR-related protein (*GITR*, or *TNFRSF18*) [28], as well as other immune checkpoint molecules [27], which is consistent with their highly immunosuppressive characteristics [28]. Ligand–receptor analysis indicated that the TIGIT–PVR pair was enriched between Tregs and malignant cells, suggesting that blocking the TIGIT–PVR axis may reduce the interaction between Tregs and malignant cells and serve as a potential therapeutic strategy for iCCAs.

Natural killer (NK) cells are a type of cytotoxic lymphocyte that is critical to the innate and adaptive immune responses. There were two NK subsets (GZMH+ and GZMK+) identified, which were mainly derived from adjacent tissues and characterized by high *GZMB*, *GZMK*, *PRF1*, and *KLRF1* expression. This indicates that these cells remained cytotoxic or activated, even when located in the adjacent tissue area. These NK cells were enriched with increased hypoxia, apoptosis, and IFN response, alongside decreased oxidative phosphorylation [28]. Notably, S100P+SPP1− iCCA had significantly reduced levels of CD56+ NK cells compared to S100P−SPP1+ iCCA, which suggests that NK cell function may differ among different TEs.

### 4.2. Myeloid Compartment

The myeloid lineage in the TE includes granulocytes, monocytes, macrophages, myeloid-derived suppressive cells, and dendritic cells (DCs). Among them, tumor-associated macrophages (TAMs) play pivotal roles in tumor progression, including initiation, promotion, immune suppression, angiogenesis, invasion, and metastasis. In iCCA, TAMs exert their immunosuppressive function likely through promoting T cell exhaustion [46], which was supported by the observation of TAMs within the tumor expressing VEGF which drove intra-tumoral heterogeneity and TE reprogramming [27]. Currently, the analysis of the myeloid lineage in the CCA TE specifically based on scRNA-seq data has been limited. A total of six heterogenous subsets in the myeloid lineage were identified based on single cell transcriptomics, including one monocyte (FCN1+), two macrophages (SPP1+ and CCL18+), and three DCs (CD1C+, XCR1+, and CD1A+). In addition, more infiltrating CD68+CD206+ macrophages in S100P+ SPP− iCCA were observed, though there was no significant difference in terms of the total CD68+ macrophage population between S100P+SPP1− and S100P−SPP1+ iCCA [30]. Macrophages and CD1a^+^ DCs were significantly enriched in tumors compared with the corresponding non-tumor tissues, while monocytes, CD1c^+^ DCs, and cDC1 DCs (XCR1^+^) showed the opposite trend. SPP1^+^ macrophages were enriched in S100P^−^SPP1^+^ iCCA, while CCL18^+^ macrophages were mostly enriched in S100P^+^ SPP^−^ iCCA. It is important to note that these two transcriptionally different macrophages in the CCA TE exhibit different functions. SPP1^+^ macrophages are skewed toward TAM1 polarization, as these cells were more potent in both the pro- and anti-inflammatory responses and exhibited increased levels of oxidative phosphorylation along with glycine, serine, threonine, and tyrosine metabolism. In contrast, CCL18^+^ macrophages showed a dominant TAM2-like phenotype with stronger tumor-promoting characteristics, elevated cytokine–cytokine receptor interaction, nitrogen and riboflavin metabolism, and high expression of CD163, MARCO, and CSF1R. Together, these results indicate the vast heterogeneity that exists within the myeloid compartment of CCA [30].

## 5. Conclusions and Future Directions

Intrinsic intra-tumoral heterogeneity in CCA has been a great challenge for the treatment of CCA. ScRNA-seq technology and associated computerized analyses have provided unprecedented insights into understanding the development of CCA. They have also allowed us to better understand transcriptomics at a higher resolution. More importantly, they have enabled researchers to illuminate the complexities of intra-tumoral heterogeneity in CCA. Now, researchers are looking to better understand the heterogeneity that exists within different cell populations in the TE, and their network of cellular interactions (Figure 1). These findings not only improve our understanding of tumor biology and mechanisms underlying therapy resistance in the case of CCA but also facilitate the exploration of novel combination therapies.

There are critical challenges that remain in addition to the well-recognized high cost and the lack of available patient samples. Firstly, CCA arises from various sites of the biliary tree, and there is a broad spectrum of mutations that are dependent on the anatomical location of the tumor. Current available scRNA-seq data is mainly derived from iCCA as there is no available data on eCCA. This there is a large need for such samples in order to fully understand the tumor biology and develop individualized therapies for CCA patients. Secondly, as stated above, a deeper analysis into the various types of immune cells in CCA is still lacking. For example, there are no data about T cell receptor sequencing for T cells in the CCA TE. It is critical to understand the clonal expansion of the unique T cell population and to search for antigen-specific T cell therapies. Nevertheless, tumor associated endothelial cells, neutrophils, and CSCs in the TE of CCA need to be better characterized to illuminate additional mechanisms that contribute to tumor initiation, metastasis, and therapy resistance. Thirdly, the scRNA-seq platform is assembled with different sensitive steps, including the necessity for high quality sample collection/preparation and library construction. Therefore, any variation during the sample preparation may change the readout and affect the final interpretation. Fourthly, scRNA-seq usually profiles only a small portion of tumor tissues and thus represents a subset of the whole cell product. Due to the technological constraints of cellular profiling, these tumor samples are obtained either through clinical needle-biopsies or surgical resection. Therefore, the results will not perfectly represent the real cellular distribution in the entire TE. Moreover, the scRNA-seq platform disrupts the original orchestrated tissue organization, and theoretical cell–cell interactions are usually interpreted with various computerized algorithms [47,48] with the cost of losing physical cell-cell contact. The absence of spatial information with scRNA-seq limits the accuracy of interpreting cell–cell interactions. The integration of scRNA-seq with other high-throughput and high-resolution single-cell technologies, e.g., spatial single-cell sequencing, single-cell proteomics, and single-cell epigenomics, will be the major advancement in understanding the fine-tuned dynamic spatial interactions between cancer cells and immune cells in CCA. Lastly, the exploration of the clinical application of scRNA-seq still remains in the hypothetical stage. However, there is tremendous hope in continuous monitoring of disease progression, evaluating therapeutic efficacy and mechanisms of resistance, developing individualized therapy, and searching for biomarkers and prognostic factors in the future after standardized procedure protocols have established guidelines on universal methods for sample harvesting, preparation, and data interpretation.

## Figures and Tables

**Figure 1 cells-12-00862-f001:**
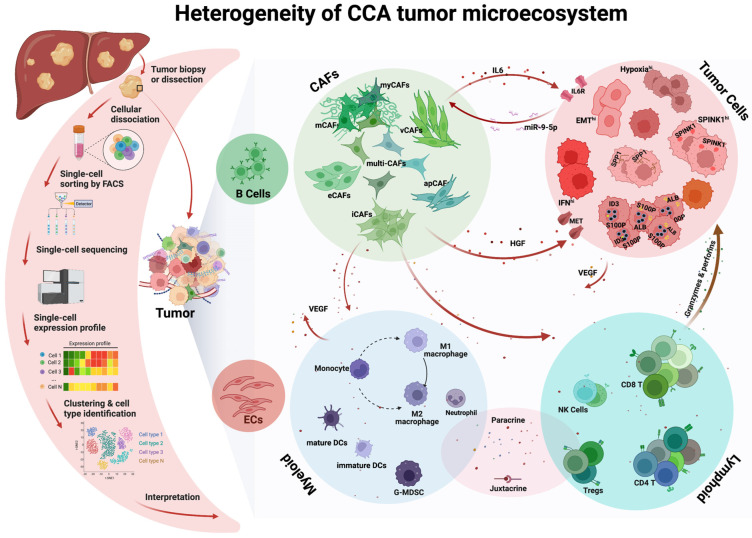
Heterogeneity of tumor ecosystem in cholangiocarcinoma defined through single cell RNA sequencing analysis. An illustration of single-cell RNA sequencing technology and heterogeneity of tumor ecosystem in cholangiocarcinoma (created with BioRender.com, accessed on 11 January2023).

**Table 1 cells-12-00862-t001:** Overview of single cell RNA sequencing studies in cholangiocarcinoma.

Species	Tumor	Tissue	Tissue Dissociation	Cells	Sequenced Cells (N)	Annotated Cell Types	Summary	Ref.
Human	9 HCC 10 iCCA	Needle biopsied	Tumor dissociation Kit (Miltenyi Biotech)	Unsorted single cell suspension	349,107	Malignant cells, CD4 T cells, CD8 T cells, B cells, CAFs, TAMs, TECs, HPC-like	Tumor transcriptomic diversity of HCC and iCCA is associated with patient outcome and T cell polarization; tumor-derived VEGF drives tumor microenvironment reprogramming.	[27]
Human	5 iCCA 3 adjacent normal tissue	Surgical resected	Dispase (Roche), type IV collagenase (Sigma), DNase I (Roche)	Unsorted single cell suspension	56,871	Malignant cells, cholangiocytes, hepatocytes, CD4 T cells, CD8 T cells, B cells, NK cells, Macrophages, DCs, fibroblasts, endothelial cells	Malignant cells display remarkable inter-tumor heterogeneity; Tregs revealed highly immunosuppressive features.	[28]
Human	2 iCCA	Surgical resected	Dispase (Roche), type IV collagenase (Sigma), DNase I (Roche)	EpCAM^−^CD45^−^CD31^−^ cells	13,150	CAFs	Six subsets are defined; CD146+ vCAFs are the most dominant fibroblasts and interact with malignant cells via IL6/IL6R axis; tumor exosomal miR-9-5p enhances IL6 expression in vCAFs and contributes to iCCA progression through EZH2.	[28]
Human	25 HCC 12 iCCA	Needle biopsied or surgical resection	Tumor dissociation Kit (Miltenyi Biotech)	Unsorted single cell suspension	91,019	Hepatocytes, cholangiocytes, CD4 T cells, CD8 T cells, B cells, CAFs, TAMs, TECs	Functional clonality of tumor cells could be a prognostic surrogate in liver cancer; tumor clonality is linked to polarized immune cell landscape and osteopontin is potential player in tumor cell evolution.	[29]
Human	14 iCCA	Surgical resected	Tumor dissociation Kit (Miltenyi Biotech)	Unsorted single cell suspension	144,878	Malignant cells, CD4 T cells, CD8 T cells, B cells, NK cells, MAIT, macrophages, DCs, monocytes, fibroblasts, endothelial cells	Two subsets of iCCA are defined with distinct marker and immune microenvironment.	[30]
Mouse	2 YAP/AKT iCCA		Collagenase D (Roche), Trypsin-EDTA (Gibco), DNase I (Roche)	Sorted Col1a1-GFP+ cells	13,026	CAFs	The majority of CAFs in iCCA are derived from HSC; iCAFs promote iCCA through HGF–MET axis; CAF-derived type I collagen contributes to stiffness but does not promote iCCA growth.	[31]
Mouse	1 YAP/AKT iCCA 1 KRAS/p19 iCCA		Collagenase D (Roche), Trypsin-EDTA (Gibco), DNase I (Roche)	Sorted Col1a1-GFP+ cells mixed with unsorted cell suspension	11,836	CAFs		
Mouse	Notch/AKT		Liver Dissociation Kit (Miltenyi Biotech)	Unsorted single cell suspension	51,897	Malignant cells, cholangiocytes, T cells, B cells, NK cells, macrophages, DCs, fibroblasts, endothelial cells, neutrophils	Stress-responding subtype and proliferating subtype of iCCA tumor cells are identified. The interaction of fibroblasts and endothelial cells promote iCCA growth.	[32]

CAFs, cancer-associated fibroblasts; DCs, dendritic cells; HPC-like, hepatic progenitor cells; HCC, hepatocellular carcinoma HGT, hepatocyte growth factor; iCAFs, inflammatory CAFs; iCCA, intrahepatic cholangiocarcinoma; MAIT, mucosal-associated invariant T cells; NK, natural killer; TAMs, tumor-associated macrophages, TECs, tumor-associated endothelial cells; Tregs, regulatory T cells; vCAFs, vascular CAFs; VEGF, vascular endothelial growth factor.

## Data Availability

Not applicable.

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
