# Peer review of "Revealing the Heterogeneity of the Tumor Ecosystem of Cholangiocarcinoma through Single-Cell Transcriptomics"

_cells, 2023, doi:10.3390/cells12060862_

Round 1

Reviewer 1 Report

The paper provides a comprehensive review of Tumor Ecosystem of Cholangiocarcinoma focusing on data obtained from single-cell RNA sequencing. The review focuses on heterogeneity and interaction between various cellular components in tumor ecosystem of cholangiocarcinoma, discusses limitations in current studies, future directions, and potential clinical applications. 

In the introduction section the authors briefly mention the actual therapeutic option for CCA; however, they do not mention anywhere that at present surgery and hepatic resection remains the primary curative choice for CCA when applicable, neither that liver transplantation is an available and probably the best curative choice for selected cases; I do think that adding these informations and some references on the topic might strengthen the paper very much.

1: Twohig P, Peeraphatdit TB, Mukherjee S. Current status of liver

transplantation for cholangiocarcinoma. World J Gastrointest Surg. 2022 Jan

27;14(1):1-11. doi: 10.4240/wjgs.v14.i1.1. PMID: 35126858; PMCID: PMC8790328.

2: Sapisochin G, Ivanics T, Heimbach J. Liver Transplantation for Intrahepatic

Cholangiocarcinoma: Ready for Prime Time? Hepatology. 2022 Feb;75(2):455-472.

doi: 10.1002/hep.32258. Epub 2022 Jan 6. PMID: 34859465.

3: Manzia TM, Parente A, Lenci I, Sensi B, Milana M, Gazia C, Signorello A,

Angelico R, Grassi G, Tisone G, Baiocchi L. Moving forward in the treatment of

cholangiocarcinoma. World J Gastrointest Oncol. 2021 Dec 15;13(12):1939-1955.

doi: 10.4251/wjgo.v13.i12.1939. PMID: 35070034; PMCID: PMC8713313.

Author Response

Thank you for comments. We have cited abovementioned three reviews as well as original studies in the introduction portion.

Reviewer 2 Report

Cholangiocarcinoma is still one of the most resistant cancers for treatment.  Authors reviewed and summarized recent understanding of tumor ecosystem of cholangiocarcinoma through single-cell transcriptomics.  Authors covered diverse topics including heterogeneity of CCA tumor cells , cancer-associated fibroblast, and immune cells. Many abbreviations were used . It would be easier for readers to follow if those abbreviations were written near the content of the topics. For example, TE, CSC. 

Author Response

Thank you for comments. The abbreviation list has been re-organized in the end of manuscript. Nevertheless, full names of some abbreviations are provided repeatedly in order to facilitate the readers to follow.

Reviewer 3 Report

The analyzed work is a narrative review of the molecular characterization of the different types of cells present in the cholangiocarcinoma tumor ecosystem through single-cell transcriptomics.

The analysis is carried out by subdividing the different cell groups present in the tumor.

In the conclusions, the critical aspect of intercellular communication from a pathogenic point of view and the possibility of treatment related to the level of manipulation of these communications are underlined.

The research work was carried out thoroughly and painstakingly, collecting data from the most up-to-date and eminent works in the scientific literature.

However, it is possible to optimize the work with a few tweaks:

- The graphics of the table are difficult to consult. Therefore it should be reformulated to allow easier access to the information entered

- You should review the list of abbreviations to enter all acronyms

- A section of "Methods" has not been inserted to insert the research criteria and the choice of the scientific sources used.

Author Response

Thank you for comments. 

1) The graphic of the table is re-organized with the deletion of one column on the left to expand the space for other columns. 2) The list of abbreviation is updated to reflect all abbreviation used in the manuscript. 3) Since this is a review rather than meta-analysis, we do not think a portion of Methods is necessary.